# Three-Dimensional Bending Analysis of Multi-Layered Orthotropic Plates by Two-Dimensional Numerical Model

**DOI:** 10.3390/ma14226959

**Published:** 2021-11-17

**Authors:** Piotr Pluciński, Jan Jaśkowiec, Maciej Wójtowicz

**Affiliations:** 1Faculty of Civil Engineering, Cracow University of Technology, Warszawska 24, 31-155 Kraków, Poland; jan.jaskowiec@pk.edu.pl; 2Faculty of Architecture, Cracow University of Technology, Warszawska 24, 31-155 Kraków, Poland; maciej.wojtowicz@pk.edu.pl

**Keywords:** multi-layered plate, three-dimensional modelling, FEM23, postprocessing

## Abstract

The paper presents effective numerical modelling of multi-layered plates with orthotropic properties. The method called the FEM23 is employed to construct the numerical model. The approach enables a full 3D analysis to be performed while using a 2D finite element mesh. The numerical model for a multi-layered plate is constructed by an assembling procedure, where each layer with orthotropic properties is added to the global numerical model. The paper demonstrates that the FEM23 method is very flexible in defining the multilayered plate, where the thickness of each layer as well as its mechanical orthotropic properties can be defined independently. Several examples of three-layered or nine-layered plates are analyzed in this paper. The results obtained by the FEM23 method coincide with the ones taken from the published papers or calculated with the standard 3D FEM approach. The orthotropic version of the FEM23 can be quite easily applied for other kinds of problems including thermo-mechanics, free vibrations, buckling analysis, or delamination.

## 1. Introduction

Multi-layer plates and shells with orthotropic properties in the layers are types of structures widely used in modern aerospace, automotive, and construction industries, due to their lightness, elasticity, durability, and excellent mechanical properties in regard to their thickness and weight. The significant increase in the use of such composite materials has been observed over the past few decades, especially in the aviation industry, where constructive components must show high strength while maintaining a slim shape. The appropriate design of structures made of orthotropic materials requires reliable computational modelling techniques.

The paper proposes a modified method called the FEM23 [1,2], whose main advantage is modelling full three-dimensional (3D) multi-layered structures while using a 2D finite element mesh for calculations. The modification of the FEM23 involves its extension to layers exhibiting orthotropic properties. The paper proves that the FEM23 method can be used effectively to analyze any multi-layered structure combining thick and thin orthotropic layers. The FEM23 is a fully 3D method, similar to the Proper Generalized Decomposition applied in [3,4], which allows for solving a fully 3D model, but with 2D characteristic computational complexity. This approach uses the spatial decomposition of a displacement field combining the in-plane 2D solution with the transverse 1D solution. As a result, another 1D mesh is generated alongside the planar 2D mesh. The method subsequently developed in [5], applies the piecewise fourth-order Lagrange polynomial along the plate’s thickness. In previous papers [6,7], the authors have already utilized the FEM23 to model the behavior of multi-layered plates and shells with isotropic layers subjected to mechanical or thermo-mechanical loads, devoting special attention to laminated glass structures—the topic of many numerical or experimental studies [8,9,10,11].

The body of research regarding the analysis of orthotropic multi-layered plates is limited here only to papers published in recent years, to show up-to-date achievements in the area. Rajaneesh et al. in [12] applied a variationally consistent new first-order shear deformation theory to three-layered orthotropic plates using an equivalent-single-layer theory in the finite element. A numerical analysis of elastic waves in the multilayered orthotropic plates was presented in the paper by Liu et al. [13], where the extended Legendre polynomials combined with the anisotropic couple-stress theory were applied to investigate the reflection and transmission of the waves. A layerwise generalized theory of layered orthotropic plates was proposed by Ugrimov and Shupicov [14]. The theory is based on a power series expansion of the displacement vector component in each layer for the transverse direction, where the number of terms retained in the power series is arbitrary and chosen according to the problem being considered. Xu et al. [15] proposed analytical solutions for orthotropic thin plates, where the finite integral transform method was introduced for accurate bending analysis. This type of analysis is limited only to thin plates due to the Kirchhoff–Love plate theory applied to derive the employed method. The vibration response of orthotropic composite plates was analysed by Zhang et al. [16] with a new approach regarding the analytical solution, while 3D semi-analytical vibrations of thick orthotropic plates were investigated by Cui et al. [17]. In this research, the modified Fourier series were applied to obtain admissible displacement functions while boundary conditions were generated by arranging sets of linear springs at the edges. Belyaev et al. [18] proposed a two-dimensional model to analyze deformations of multi-layered orthotropic plates, where a multi-layered plate was replaced by an equivalent plate composed of a monoclinic material with piecewise elastic modules, and an asymptotic expansion was applied to obtain the solution. The buckling analysis of orthotropic laminated plates with asymmetrical boundary conditions was carried out by Schreiber et al. [19]. The framework of Reddy’s third-order shear deformation theory was applied for a closed-form solution, and Lévy-type solution was derived for the plate buckling problem.

The multi-layered and orthotropic plates are the subject of both numerical and experimental research. The combined experimental and analytical study of multi-layered laminated glass subjected to low-velocity impacts was presented by Wang et al. [20], while the experimental determination of material properties in an elliptical orthotropic plate was proposed by Marchetti et al. [21]. The method validated subsequently in a honeycomb sandwich panel combined the equivalent thin plate theory with the wave fitting approach. An innovative transparent polymeric laminate consisting of two poly(methyl methacrylate) plies with a thermoplastic polyurethane interlayer was experimentally and numerically investigated by Rühl et al. [22]. The laminate was subjected to low-velocity impact loadings using clamped three-point bending and dart impact tests at different temperatures and velocities. Multi-layered metallic plates were the subject of a number of experiments conducted by Ziya-Shamami et al. [23]. The paper analyzed 15 various testing groups, presenting the data obtained in over a hundred experiments. The plates were subjected to repeated uniform impulsive loadings to study the effects of different charge masses, layering configurations, layering arrangements, layer thicknesses, and stand-off distances on the central deflection of single-, double-, and triple-layered mixed plates made of aluminum alloy and mild steel materials.

In this paper, the FEM23 method is extended in such a way that a multi-layered plate with orthotropic properties can be analyzed. The orthotropy in the numerical model is obtained by the stress–strain constitutive relation defined in the coordinates associated with the orthotropy directions, subsequently transformed to global coordinates and applied to the mathematical model shown in detail in Section 2. A specific form of displacement approximation used for the purpose of this study and the final numerical model are presented in Section 3. The FEM23 approach for orthotropic multi-layered plates is verified in three examples analyzed in Section 4. The paper ends with some conclusions.

## 2. Variational Formulation of the Problem

The starting point of the presented FEM23 method for the multi-layered orthotropic plates is a single-layered orthotropic plate model subsequently extended to a multi-layered one. Two Cartesian coordinates, orthotropic (ξ,η,θ) and global (x,y,z), are distinguished in the orthotropic plate. The θ and *z* directions are the same and oriented transversely, while the (ξ,η) pair is associated with in-plane orthotropy in a layer, and the (x,y) pair is correlated with the in-plane geometry of the plate. Although the (x,y) pair is common for the whole plate, the (ξ,η) directions are oriented differently in each layer. The orthotropic coordinates are used to derive constitutive relations in the layers, while the global ones are used to describe the model for the whole multi-layered plate.

The constitutive (Hooke’s) relation for orthotropic materials is expressed with the orthotropic coordinates as follows:(1)$$\begin{array}{c} \tilde{\sigma }\left(\tilde{u}\right)=\tilde{D}:\tilde{\varepsilon }\left(\tilde{u}\right), \end{array}$$
where $\tilde{u}$ is the displacement vector, while $\tilde{\sigma }$ and $\tilde{\varepsilon }$ are the stress and Cauchy’s strain tensors, all defined with the use of the orthotropic coordinates. $\tilde{D}$ is a symmetric constitutive tensor of the fourth order in the orthotropic coordinates, demonstrating the following non-zero values [24]:$$\begin{array}{cc} \tilde{D}{}_{1111} & =\left(1-\nu_{23}\nu_{32}\right)E_{1}\gamma ={\bar{D}}_{1};\\ \tilde{D}{}_{2222} & =\left(1-\nu_{13}\nu_{31}\right)E_{2}\gamma ={\bar{D}}_{2};\\ \tilde{D}{}_{3333} & =\left(1-\nu_{12}\nu_{21}\right)E_{3}\gamma ={\bar{D}}_{3};\\ \tilde{D}{}_{2233} & =\tilde{D}{}_{3322}=\left(\nu_{32}+\nu_{12}\nu_{31}\right)E_{2}\gamma =\left(\nu_{23}+\nu_{21}\nu_{13}\right)E_{3}\gamma ={\bar{D}}_{4};\\ \tilde{D}{}_{1133} & =\tilde{D}{}_{3311}=\left(\nu_{31}+\nu_{21}\nu_{32}\right)E_{1}\gamma =\left(\nu_{13}+\nu_{12}\nu_{23}\right)E_{3}\gamma ={\bar{D}}_{5};\\ \tilde{D}{}_{1122} & =\tilde{D}{}_{2211}=\left(\nu_{21}+\nu_{31}\nu_{23}\right)E_{1}\gamma =\left(\nu_{12}+\nu_{13}\nu_{32}\right)E_{2}\gamma ={\bar{D}}_{6};\\ \tilde{D}{}_{2323} & =\tilde{D}{}_{3232}=\tilde{D}{}_{2332}=\tilde{D}{}_{3223}=G_{23}={\bar{D}}_{7};\\ \tilde{D}{}_{1313} & =\tilde{D}{}_{3131}=\tilde{D}{}_{1331}=\tilde{D}{}_{3113}=G_{13}={\bar{D}}_{8};\\ \tilde{D}{}_{1212} & =\tilde{D}{}_{2121}=\tilde{D}{}_{1221}=\tilde{D}{}_{2112}=G_{12}={\bar{D}}_{9}, \end{array}$$
where
$$\begin{array}{c} \gamma =\frac{1}{1-\nu_{12}\nu_{21}-\nu_{23}\nu_{32}-\nu_{31}\nu_{13}-2\nu_{21}\nu_{32}\nu_{13}} \end{array}$$

$E_{i}$ represents the Young moduli of elasticity, $G_{ij}$ is shear moduli, while $\nu_{ij}$ are Poisson ratios, all in orthotropic directions. Using the symmetric properties of the $\tilde{D}$ tensor, the following relation is derived:(2)$$\begin{array}{c} \nu_{ij}=\nu_{ji}E_{i}/E_{j} \end{array}$$

The stress and strain tensors from Equation (Equation 1) satisfy the equilibrium equation, which can be written in the variational form for the whole considered domain Ω:(3)$$\begin{array}{c} \begin{array}{cc} \; & \int_{\Gamma^{\sigma }}\tilde{v}\cdot \hat{\tilde{t}}\;d\Gamma -\int_{\Omega }\tilde{\varepsilon }\left(\tilde{v}\right):\tilde{D}:\tilde{\varepsilon }\left(\tilde{u}\right)+\int_{\Omega }\tilde{v}\cdot \tilde{b}\;d\Omega =0\\ & \tilde{u}=\hat{\tilde{u}}\;\mathrm{on}\;\Gamma^{u}, \end{array} \end{array}$$
which has to be satisfied for all $\tilde{v}$ where $\tilde{v}=0$ on $\Gamma^{u}$.

The second integral in Equation (Equation 3) refers to the tensor inner product. For the sake of further analysis, it can be shown that, due to the symmetric properties of the $\tilde{D}$ tensor, the inner product can be expressed with full displacement gradients:(4)$$\begin{array}{c} \varepsilon \left(\tilde{v}\right):\tilde{D}:\varepsilon \left(\tilde{u}\right)=\tilde{\nabla }{\tilde{v}}^{T}:\tilde{D}:\tilde{\nabla }\tilde{u}, \end{array}$$
where $\tilde{\nabla }$ is the gradient operator defined with the orthotropic coordinates.

Assuming that Q is the transformation matrix from the orthotropic to global coordinates, the relation between the displacement gradients expressed with the orthotropic and global coordinates is as follows:(5)$$\begin{array}{c} \begin{array}{c} \tilde{\nabla }\tilde{u}=Q^{T}\nabla uQ, \end{array} \end{array}$$
where u is the displacement vector in the global coordinates, and **∇** is the gradient operator in the global coordinates. The displacement gradient is defined in this paper as
(6)$$\begin{array}{c} \nabla u=\left[\begin{array}{ccc} \frac{\partial u_{x}}{\partial x} & \frac{\partial u_{y}}{\partial x} & \frac{\partial u_{z}}{\partial x}\\ \frac{\partial u_{x}}{\partial y} & \frac{\partial u_{y}}{\partial y} & \frac{\partial u_{z}}{\partial y}\\ \frac{\partial u_{x}}{\partial z} & \frac{\partial u_{y}}{\partial z} & \frac{\partial u_{z}}{\partial z} \end{array}\right] \end{array}$$

When the relation in Equation (Equation 5) is subsequently applied to the inner product in Equation (Equation 4), the following relations are derived:(7)$$\begin{array}{c} \tilde{\nabla }{\tilde{v}}^{T}:\tilde{D}:\tilde{\nabla }\tilde{u}=Q^{T}\nabla v^{T}Q:\tilde{D}:Q^{T}uQ=\nabla v^{T}:D:\nabla u, \end{array}$$
where D is the constitutive tensor written in the global coordinates with its components expressed as
(8)$$\begin{array}{c} D_{ijkl}=\sum_{nmpr}Q_{in}Q_{jm}Q_{kp}Q_{lr}{\tilde{D}}_{nmpr} \end{array}$$

In this paper, the orthotropic and global plate coordinates are always rotated around the *z* axis. Consequently, the transformation matrix Q takes the following form:(9)$$\begin{array}{c} Q=\left[\begin{array}{ccc} c & s & 0\\ -s & c & 0\\ 0 & 0 & 1 \end{array}\right], \end{array}$$
where c=cos(α), s=sin(α) and α constitute a rotation angle between the first axes of the global and orthotropic coordinate systems around the *z* axis. The relation between those two coordinates is illustrated in Figure 1.

Taking into account the form of the transformation matrix shown in Equation (Equation 9) the Hook tensor defined with the global coordinates in Equation (Equation 8), takes the following form:$$\begin{array}{cc} D{}_{1111} & ={\bar{D}}_{1}c^{4}+{\bar{D}}_{2}s^{4}+\left(2{\bar{D}}_{6}+4{\bar{D}}_{9}\right)c^{2}s^{2}=D_{1};\\ D{}_{2222} & ={\bar{D}}_{1}s^{4}+{\bar{D}}_{2}c^{4}+\left(2{\bar{D}}_{6}+4{\bar{D}}_{9}\right)c^{2}s^{2}=D_{2};\\ D{}_{3333} & ={\bar{D}}_{3}=D_{3};\\ D{}_{2233} & =D{}_{3322}={\bar{D}}_{4}c^{2}+{\bar{D}}_{5}s^{2}=D_{4};\\ D{}_{1133} & =D{}_{3311}={\bar{D}}_{4}s^{2}+{\bar{D}}_{5}c^{2}=D_{5};\\ D{}_{1122} & =D{}_{2211}=\left({\bar{D}}_{1}+{\bar{D}}_{2}-4{\bar{D}}_{9}\right)c^{2}s^{2}+{\bar{D}}_{6}\left(c^{4}+s^{4}\right)=D_{6};\\ D{}_{2323} & =D{}_{3232}=D{}_{2332}=D{}_{3223}={\bar{D}}_{7}c^{2}+{\bar{D}}_{8}s^{2}=D_{7};\\ D{}_{1313} & =D{}_{3131}=D{}_{1331}=D{}_{3113}={\bar{D}}_{7}s^{2}+{\bar{D}}_{8}c^{2}=D_{8};\\ D{}_{1212} & =D{}_{2121}=D{}_{1221}=D{}_{2112}=\left({\bar{D}}_{1}+{\bar{D}}_{2}-2{\bar{D}}_{6}\right)c^{2}s^{2}+{\bar{D}}_{9}{\left(c^{2}-s^{2}\right)}^{2}=D_{9};\\ D{}_{1332} & =D{}_{3132}=D{}_{1323}=D{}_{3123}=D{}_{2313}=D{}_{3213}=D{}_{2331}=D{}_{3231}=\left({\bar{D}}_{7}-{\bar{D}}_{8}\right)cs=D_{10};\\ D{}_{1233} & =D{}_{2133}=D{}_{3312}=D{}_{3321}=\left({\bar{D}}_{4}-{\bar{D}}_{5}\right)cs=D_{11};\\ D{}_{1222} & =D{}_{2122}=D{}_{2212}=D{}_{2221}={\bar{D}}_{2}c^{3}s-{\bar{D}}_{1}cs^{3}+\left({\bar{D}}_{6}+2{\bar{D}}_{9}\right)\left(cs^{3}-c^{3}s\right)=D_{12};\\ D{}_{1112} & =D{}_{1121}=D{}_{1211}=D{}_{2111}={\bar{D}}_{2}cs^{3}-{\bar{D}}_{1}c^{3}s+\left({\bar{D}}_{6}+2{\bar{D}}_{9}\right)\left(c^{3}s-cs^{3}\right)=D_{13} \end{array}$$

The integral Equation (Equation 3) expressed with the orthotropic coordinates can now be rewritten, so that all the components are described with the global coordinates
(10)$$\begin{array}{c} \begin{array}{cc} \; & \int_{\Gamma^{\sigma }}v\cdot \hat{t}\;d\Gamma -\int_{\Omega }\varepsilon \left(v\right):D:\varepsilon \left(u\right)+\int_{\Omega }v\cdot b\;d\Omega =0\\ & u=\hat{u}\;\mathrm{on}\;\Gamma^{u} \end{array} \end{array}$$

Using the method described in the authors’ previous papers, for instance, [1,2,6], the inner product in Equation (Equation 7) is then expressed with the derivatives in the transverse and in-plane directions, consequently changing into
(11)$$\begin{array}{c} \begin{array}{cc} \varepsilon (v):D:\varepsilon (u) & =\frac{\partial v}{\partial z}\cdot D_{1}\cdot \frac{\partial u}{\partial z}+\frac{\partial v}{\partial z}\cdot D_{2}:\bar{\nabla }u+\bar{\nabla }v^{T}:D_{3}\cdot \frac{\partial u}{\partial z}+\bar{\nabla }v^{T}:D_{4}:\bar{\nabla }u, \end{array} \end{array}$$
where
(12)$$\begin{array}{c} \bar{\nabla }u=\left[\begin{array}{ccc} \frac{\partial u_{x}}{\partial x} & \frac{\partial u_{y}}{\partial x} & \frac{\partial u_{z}}{\partial x}\\ \frac{\partial u_{x}}{\partial y} & \frac{\partial u_{y}}{\partial y} & \frac{\partial u_{z}}{\partial y} \end{array}\right] \end{array}$$

The $D_{i}$ tensors constitute adequate parts of the Hooke’s D tensor, defined as follows:(13)$$\begin{array}{c} \begin{array}{cc} \; & {\left(D_{1}\right)}_{ij}=D_{i33j}\;,\;i=1,2,3\;j=1,2,3;\\ & {\left(D_{2}\right)}_{ijk}=D_{i3jk}\;,\;i=1,2,3\;j=1,2\;k=1,2,3;\\ & {\left(D_{3}\right)}_{ijk}=D_{ij3k}\;,\;i=1,2,3\;j=1,2\;k=1,2,3;\\ & {\left(D_{4}\right)}_{ijkl}=D_{ijkl}\;,\;i=1,2,3\;j=1,2\;k=1,2\;l=1,2,3 \end{array} \end{array}$$

The relations in Equation (Equation 13) provide definitions of $D_{i}$ tensors in the relation D one. However, the explicit definitions of the non-zero components of the tensors are as follows
(14)$$\begin{array}{c} \begin{array}{cc} \; & {\left(D_{1}\right)}_{11}=D_{8}\;,\;{\left(D_{1}\right)}_{22}=D_{7}\;,\;{\left(D_{1}\right)}_{33}=D_{3}\\ & {\left(D_{1}\right)}_{12}={\left(D_{1}\right)}_{21}=D_{10} \end{array} \end{array}$$
(15)$$\begin{array}{c} \begin{array}{cc} \; & {\left(D_{2}\right)}_{131}=D_{8}\;,\;{\left(D_{2}\right)}_{132}={\left(D_{2}\right)}_{231}=D_{10}\;,\;{\left(D_{2}\right)}_{232}=D_{7}\\ \; & {\left(D_{2}\right)}_{311}=D_{5}\;,\;{\left(D_{2}\right)}_{312}={\left(D_{2}\right)}_{321}=D_{11}\;,\;{\left(D_{2}\right)}_{322}=D_{4} \end{array} \end{array}$$
(16)$$\begin{array}{c} D_{3}=D_{2}^{T}\;\Rightarrow \;{\left(D_{3}\right)}_{ijk}={\left(D_{2}\right)}_{kji} \end{array}$$
(17)$$\begin{array}{c} \begin{array}{cc} \; & {\left(D_{4}\right)}_{1111}=D_{1},\;{\left(D_{4}\right)}_{2222}=D_{2},\;{\left(D_{4}\right)}_{1331}=D_{8},\;{\left(D_{4}\right)}_{2332}=D_{7}\\ \; & {\left(D_{4}\right)}_{1122}={\left(D_{4}\right)}_{2211}=D_{6},\;{\left(D_{4}\right)}_{2331}={\left(D_{4}\right)}_{1332}=D_{10}\\ \; & {\left(D_{4}\right)}_{1212}={\left(D_{4}\right)}_{1221}={\left(D_{4}\right)}_{2121}={\left(D_{4}\right)}_{2112}=D_{9}\\ \; & {\left(D_{4}\right)}_{1222}={\left(D_{4}\right)}_{2122}={\left(D_{4}\right)}_{2212}={\left(D_{4}\right)}_{2221}=D_{12}\\ \; & {\left(D_{4}\right)}_{2111}={\left(D_{4}\right)}_{1211}={\left(D_{4}\right)}_{1121}={\left(D_{4}\right)}_{1112}=D_{13} \end{array} \end{array}$$

Equation (Equation 11) is now applied to Equation (Equation 10). Moreover, the volume integrals are separated into the in-plane and transverse ones: (18)$$\begin{array}{c} \begin{array}{cc} \; & \int_{\Gamma^{m}}\int_{-\frac{h}{2}}^{\frac{h}{2}}\frac{\partial v}{\partial z}\cdot D_{1}\cdot \frac{\partial u}{\partial z}\;dz\;d\Gamma +\int_{\Gamma^{m}}\int_{-\frac{h}{2}}^{\frac{h}{2}}\left(\frac{\partial v}{\partial z}\cdot D_{2}:\bar{\nabla }u+\bar{\nabla }v^{T}:D_{3}\cdot \frac{\partial u}{\partial z}\right)\;dz\;d\Gamma \\ & +\int_{\Gamma^{m}}\int_{-\frac{h}{2}}^{\frac{h}{2}}\bar{\nabla }v^{T}:D_{4}:\bar{\nabla }u\;dz\;d\Gamma =\int_{\Gamma^{m}}\int_{-\frac{h}{2}}^{\frac{h}{2}}v\cdot b\;dz\;d\Gamma +\int_{\Gamma^{\sigma }}v\cdot \hat{t}\;d\Gamma \end{array} \end{array}$$
where $\Gamma^{m}$ is the mid-surface of the plate. The 2D integration over the $\Gamma^{m}$ surface can be performed using a 2D finite element mesh, while the transverse one-dimensional integration can be completed using standard 1D numerical integration schemes
(19)$$\begin{array}{c} \begin{array}{cc} \; & \sum_{i}^{n_{g}}\int_{\Gamma^{m}}\left(\frac{\partial v}{\partial z}\cdot D_{1}\cdot \frac{\partial u}{\partial z}\right){|}_{z=z_{i}}w_{i}\;d\Gamma +\sum_{i}^{n_{g}}\int_{\Gamma^{m}}\left(\frac{\partial v}{\partial z}\cdot D_{2}:\bar{\nabla }u+\bar{\nabla }v^{T}:D_{3}\cdot \frac{\partial u}{\partial z}\right){|}_{z=z_{i}}w_{i}\;d\Gamma \\ & +\sum_{i}^{n_{g}}\int_{\Gamma^{m}}\left(\bar{\nabla }v^{T}:D_{4}:\bar{\nabla }u\right){|}_{z=z_{i}}w_{i}\;d\Gamma =\sum_{i}^{n_{g}}\int_{\Gamma^{m}}v\cdot b\;d{|}_{z=z_{i}}w_{i}\;d\Gamma +\int_{\Gamma^{\sigma }}v\cdot \hat{t}\;d\Gamma \end{array} \end{array}$$
where $z_{i}$ are the integration points in the transverse direction.

Equation (Equation 19) is written for a single-layered orthotropic plate. Assuming that the plate comprises *M* orthotropic layers, each with its own orthogonal orientation, the variational equation for the whole plate is as follows:(20)$$\begin{array}{c} \begin{array}{cc} \; & \sum_{j}^{M}\sum_{i}^{n_{g}^{j}}\int_{\Gamma^{m}}\left(\frac{\partial v}{\partial z}\cdot D_{1}^{j}\cdot \frac{\partial u}{\partial z}\right){|}_{z=z_{i}^{j}}w_{i}^{j}\;d\Gamma +\sum_{j}^{M}\sum_{i}^{n_{g}^{j}}\int_{\Gamma^{m}}\left(\frac{\partial v}{\partial z}\cdot D_{2}^{j}:\bar{\nabla }u+\bar{\nabla }v^{T}:D_{3}^{j}\cdot \frac{\partial u}{\partial z}\right){|}_{z=z_{i}^{j}}w_{i}^{j}\;d\Gamma \\ & +\sum_{j}^{M}\sum_{i}^{n_{g}^{j}}\int_{\Gamma^{m}}\left(\bar{\nabla }v^{T}:D_{4}^{j}:\bar{\nabla }u\right){|}_{z=z_{i}^{j}}w_{i}^{j}\;d\Gamma =\sum_{j}^{M}\sum_{i}^{n_{g}^{j}}\int_{\Gamma^{m}}v\cdot b{|}_{z=z_{i}^{j}}w_{i}^{j}\;d\Gamma +\int_{\Gamma^{\sigma }}v\cdot \hat{t}\;d\Gamma , \end{array} \end{array}$$
where $D_{k}^{j}$ are material tensors defined for the *j*th layer of the plate and $z_{i}^{j}$ are the transverse integration points for the *j*th layer.

## 3. Approximations

The details concerning spatial approximation have already been provided by Jaśkowiec et al. [1]. This paper presents only an outline of the approximation scheme which is derived first for a single layer and then extended to a multi-layered case.

The spatial approximation of the displacement in the *j*th layer is a combination of the in-plane approximation and the transverse one. Using the 1D transverse approximation along the plate’s thickness, the displacement vector $u^{\left(j\right)}\left(x,y,z\right)$ is expressed in the following way:(21)$$\begin{array}{cc} u^{\left(j\right)}\left(x,y,z\right) & =\sum_{i=1}^{S_{j}}N_{i}\left(z\right)u^{(j)i}\left(x,y\right), \end{array}$$
where $S_{j}$ is the number of approximation surfaces specified along the thickness of the *j*th layer dependent on the transverse approximation order, $N_{i}$ is the *i*-th Lagrange polynomial of the appropriate $S_{j}-1$ order, while $u^{(j)i}\left(x,y\right)$ is the displacement vector on the *i*-th approximation surface of the *j*th layer. The same kind of the displacement decomposition has been previously proposed in [25,26] for hierarchical shell models. The approximation order depends on the layer’s thickness. The first order of the transverse approximation may be applied when the layer is thin. However, higher orders of the transverse approximation have to be utilized when the thickness of the layer is significant in relation to the outer dimensions of the plate. On the other hand, every vector $u^{i}\left(x,y\right)$ is, as in previous papers such as [1,25,26,27,28], approximated using the same approximation field constructed on a single 2D mesh:(22)$$\begin{array}{c} u^{(j)i}\left(x,y\right)=\Phi \left(x,y\right){\overset{ˇ}{u}}^{(j)i}, \end{array}$$
where Φ(x,y) is the approximation matrix defined on a 2D in-plane mesh, and ${\overset{ˇ}{u}}^{(j)i}$ is the vector of nodal displacements in the *i*-th approximation surface of the *j* layer. Using the Equations (Equation 21) and (Equation 22), the displacement vector in the entire *j*th layer is approximated as follows:(23)$$\begin{array}{c} \begin{array}{cc} u^{\left(j\right)}\left(x,y,z\right) & =\left[\begin{array}{cccc} N_{1}\left(z\right)\Phi \left(x,y\right) & N_{2}\left(z\right)\Phi \left(x,y\right) & \ldots & N_{nS}\left(z\right)\Phi \left(x,y\right) \end{array}\right]\overset{ˇ}{u}\\ & =\Psi \left(x,y,z\right){\overset{ˇ}{u}}^{\left(j\right)}, \end{array} \end{array}$$
where the global vector of degrees of freedom ${\overset{ˇ}{u}}^{\left(j\right)}$ comprises the vectors associated with the subsequent surfaces of the *j* layer
(24)$$\begin{array}{c} {\overset{ˇ}{u}}^{\left(j\right)}=\left[\begin{array}{c} {\overset{ˇ}{u}}^{(j)1}\\ {\overset{ˇ}{u}}^{(j)2}\\ \vdots \\ {\overset{ˇ}{u}}^{\left(j\right)S_{j}} \end{array}\right] \end{array}$$

Two consecutive, adjacent layers share a common surface and, consequently, the degrees of freedom associated with it. The common degrees of freedom of the two adjacent layers satisfy the relation
(25)$$\begin{array}{c} {\overset{ˇ}{u}}^{\left(j\right)S_{j}}={\overset{ˇ}{u}}^{(j+1)1} \end{array}$$

The variational Equation (Equation 20) requires the in-plane gradient and the transverse derivative of the displacement field. Such derivatives for the displacement approximation in Equation (Equation 23) are expressed as follows:(26)$$\begin{array}{c} \begin{array}{cc} \bar{\nabla }u^{\left(j\right)}\left(x,y,z\right) & =\left[\begin{array}{cccc} N_{1}\left(z\right)\bar{\nabla }\Phi \left(x,y\right) & N_{2}\left(z\right)\bar{\nabla }\Phi \left(x,y\right) & \ldots & N_{nS}\left(z\right)\bar{\nabla }\Phi \left(x,y\right) \end{array}\right]{\overset{ˇ}{u}}^{\left(j\right)}\\ & =\bar{\nabla }\Psi \left(x,y,z\right){\overset{ˇ}{u}}^{\left(j\right)}\\ \frac{\partial u^{\left(j\right)}}{\partial z}\left(x,y,z\right) & =\left[\begin{array}{cccc} N_{1}^{\prime }\left(z\right)\Phi \left(x,y\right) & N_{2}^{\prime }\left(z\right)\Phi \left(x,y\right) & \ldots & N_{nS}^{\prime }\left(z\right)\Phi \left(x,y\right) \end{array}\right]{\overset{ˇ}{u}}^{\left(j\right)}\\ & =\Psi {,}_{z}\left(x,y,z\right){\overset{ˇ}{u}}^{\left(j\right)} \end{array} \end{array}$$

After substituting the approximation shown in Equation (Equation 23) and derivatives from Equation (Equation 26) to Equation (Equation 20), the following discrete system of the equation is obtained:(27)$$\begin{array}{c} \left(K_{1}+K_{2}+K_{3}+K_{4}\right)\overset{ˇ}{u}=F, \end{array}$$
where particular matrices and the right-hand-side vector are generated with the assembling procedure across the plate’s layers
(28)$$\begin{array}{c} \overset{ˇ}{u}=\sum_{j}^{M}{A^{\left(j\right)}}^{T}{\overset{ˇ}{u}}^{\left(j\right)}\;,\;K_{n}=\sum_{j}^{M}{A^{\left(j\right)}}^{T}K_{n}^{\left(j\right)}A^{\left(j\right)}\;,\;F=\sum_{j}^{M}{A^{\left(j\right)}}^{T}F_{b}^{\left(j\right)}+{A^{\left(\sigma \right)}}^{T}F_{t}^{\left(\sigma \right)}, \end{array}$$
where $A^{\left(j\right)}$ is the assembling operator for the *j*th layer. The definitions of other components in Equation (Equation 28) are as follows: (29)$$\begin{array}{c} \begin{array}{cc} \; & K_{1}^{\left(j\right)}=\sum_{i}^{n_{g}^{j}}w_{i}^{j}\int_{\Gamma^{m}}\Psi {,}_{z}\left(x,y,z_{i}^{j}\right)\cdot D_{1}\cdot \Psi {,}_{z}\left(x,y,z_{i}^{j}\right)\;d\Gamma ;\\ \; & K_{2}^{\left(j\right)}=\sum_{i}^{n_{g}^{j}}w_{i}^{j}\int_{\Gamma^{m}}\Psi {,}_{z}\left(x,y,z_{i}^{j}\right)\cdot D_{2}:\bar{\nabla }\Psi \left(x,y,z_{i}^{j}\right)\;d\Gamma ,\;K_{3}^{\left(j\right)}={K_{2}^{\left(j\right)}}^{T};\\ \; & K_{4}^{\left(j\right)}=\sum_{i}^{n_{g}^{j}}w_{i}^{j}\int_{\Gamma^{m}}\bar{\nabla }\Psi^{T}\left(x,y,z_{i}^{j}\right):D_{4}:\bar{\nabla }\Psi \left(x,y,z_{i}^{j}\right)\;d\Gamma ;\\ & F_{b}^{\left(j\right)}=\sum_{i}^{n_{g}^{j}}w_{i}^{j}\int_{\Gamma^{m}}\Psi^{T}(x,y,z_{i}^{j})\cdot b\;d\Gamma \;,\;F_{t}^{\left(\sigma \right)}=\int_{\Gamma^{\sigma }}\Psi^{T}\left(x,y,z_{\sigma }\right)\cdot \hat{t}\;d\Gamma \end{array} \end{array}$$
where $z_{\sigma }$ is the surface on the top layer, where traction $\hat{t}$ is applied.

## 4. Examples

This section presents three examples demonstrating the accuracy and efficiency of the FEM23 method for orthotropic multi-layered plates. In the examples, the results yielded by the FEM23 approach are compared with the ones from other published study. The first example, Section 4.1, focuses on a simply supported three-layered square plate whose layers demonstrate the same mechanical properties, but their orthotropic directions are rotated. The results yielded with the layerwise theory and 3D FEM are compared. Additionally, various types of simply supported boundary conditions and the case study of plates with varying thickness are analyzed. The second example, Section 4.2, investigates a plate with layers of varying values of thickness. In the first case, the plate consists of three layers, and in the second one, of nine layers constructed with three different materials. This example shows how the FEM23 method copes with multi-layered structures made of different orthotropic materials with various thickness. The FEM23 results are verified with the zigzag theory and 3D FEM. In the third example, the plate under consideration is a three-layered one, in which the orthotropic directions are rotated in relation to each other. In this example, the FEM23 results are again compared with the ones obtained using the zigzag theory and the standard 3D FEM.

### 4.1. Example 1

In this example, a simply supported three-layered square plate loaded by the pressure on the upper outer surface is analyzed. The side length of the plate is *a*, while the thickness of the plate is *h* with $\frac{h}{3}$ thickness for each layer. The geometric and material properties of the plate are shown in Figure 2. In this example, the results obtained by the FEM23 are compared with the ones presented in [29,30], where layerwise plate theories were applied. In the FEM23 a full 3D analysis is performed, using standard 3D finite elements in the ABAQUS application.

The fact that the plate is simply supported can be easily applied in the case of plate theory. However, in a 3D analysis, such boundary conditions are ambiguous. In a simply supported 3D plate, the outer points at the bottom of the plate may be blocked and in another approach—the outer points at the mid-surface of the plate. Moreover, the plate can be hard or simply supported. Figure 3 presents graphically different versions of the simply supported boundary conditions in a 3D analysis.

In this example, the results obtained by the FEM23 method are compared with the ones taken from the papers by Vuksanović et al. [30] and Carrera and Ciuffreda [29], where multilayer plate theories were applied, as well as with the results yielded with the standard 3D FEM method using the Abaqus package. In the FEM approach, each layer of the plate is discretized by a single layer of hexahedral elements. Table 1 shows all the results in the form of a dimensionless deflection ${\bar{u}}_{z}$ and stress component ${\bar{\sigma }}_{x}$, both evaluated at two central points of the plate:
(30a)$$\begin{array}{cc} \; & {\bar{u}}_{z}=\frac{E_{2}100h^{3}}{q_{0}a^{4}}u_{z}\left(\frac{a}{2},\frac{a}{2},0\right) \end{array}$$
(30b)$$\begin{array}{cc} & {\bar{\sigma }}_{x}={\left(\frac{h}{a}\right)}^{2}\frac{1}{q_{0}}\sigma_{x}\left(\frac{a}{2},\frac{a}{2},-\frac{h}{2}\right) \end{array}$$

The results yielded with the FEM23 and FEM presented in Table 1 coincide for all sorts of applied supports and for every plate thickness, which proves that the FEM23 provides an accurate 3D analysis. The results are also in good correlation with the ones obtained with plate theories, but only when boundary conditions are applied to the mid-surface nodes in the FEM23 and a thin plate is considered. When the relation a/h increases, both types of results become inconsistent, which is natural for plate models based on the Kirchhoff–-Love theory efficient only for thin plates.

The results are illustrated with the plots of the dimensionless deflections and the stress ${\bar{\sigma }}_{x}$ for the plates with a/h=4 and a/h=100 along the plate’s thickness shown in the middle of the plate (see Figure 4 and Figure 5). It can be noted that the application of the boundary conditions plays a significant part in the case of the 3D model. Different results are yielded when the support is applied at the bottom edge or in the middle of the plate’s thickness. The bottom edge support case is very sensitive to both hard and soft application of the boundary condition. Such sensitivity is not observed when the middle edge support is used. The following Figures show the comparison of the obtained results with the ones based on the plate theory taken from [29]. In the case of thin plate, the results of the [mh] type boundary conditions almost overlap (see Figure 5). However, when a thick plate is considered (see Figure 4), these two types of results differ, which is typical for the plate theory approaches.

The FEM23 method uses full 3D modelling with the calculations performed on a 2D domain. In this example, the FEM23 results are compared with the ones obtained by the standard 3D FEM method with Abaqus application. As can be seen in Table 1, both approaches yield very similar results for all types of boundary conditions. Both methods are also compared in Figure 6, where the presented 3D deflection and stress maps are very similar to each other. It can be observed that the relative differences of the maximum deflections and stresses are around $7\cdot 10^{-3}$ and $2\cdot 10^{-1}$, respectively.

### 4.2. Example 2

The following example, excerpted from [31], applying the Hellinger–Reissner refined zigzag theory (HR-RZT), analyzes a square multi-layered plate of a=2 m total thickness h= 0.1 m], with a constant load $q_{0}$ = 0.1 N/mm${}^{2}$. The plate is constructed of three types of materials called A, B, and C, whose properties are provided in Table 2. The material A is an orthotropic carbon-fiber reinforced plastic, the material B represents an orthotropic titanium honeycomb structure, and the material C corresponds to an isotropic foam made of polyvinyl chloride.

The analysis involves, as in considered paper [31], two types of multi-layered plates: the ones consisting of three and nine layers. The structures of the two plates are presented in Table 3, and the nine-layered plate is shown in Figure 7. It is assumed that the angle of the orthotropic direction α is equal to zero for each layer. The calculations performed in the FEM23 method utilize a quadrilateral mesh structure with 42 × 42 nine-node elements. In this example, the coordinates’ origin is located in the middle of the plate. The boundary conditions are as follows:(31)$$\begin{array}{c} \begin{array}{cc} \; & u_{z}=0\;\mathrm{for}\;\mathrm{the}\;\mathrm{points}\;\left(\pm a/2,y,0\right)\;\mathrm{and}\;\left(x,\pm a/2,0\right);\\ & u_{x}=u_{y}=0\;\mathrm{for}\;\mathrm{the}\;\mathrm{point}\;\left(-a/2,-a/2,0\right);\\ & u_{y}=0\;\mathrm{for}\;\mathrm{the}\;\mathrm{point}\;\left(a/2,-a/2,0\right) \end{array} \end{array}$$

The results presented in the form of diagrams of deflection and selected quantities along the plate’s thickness are shown in Figure 8 and Figure 9 for the L1 and L2 plates, respectively. The diagrams provide a comparison of the FEM23 results with the ones yielded by the HR-RZT and full 3D FEM approaches provided in the paper [31]. In the case of the three-layered plate L1, all three types of results coincide. When the case of the nine-layered L2 plate is considered, the results of the 3D FEM and FEM23 are almost the same and differ only slightly when compared with the HR-RZT results. All the above observations confirm the effectiveness of the FEM23 approach for a numerical analysis of multi-layered phototropic plates.

### 4.3. Example 3

The following example investigates a three-layered 50 × 50 mm composite square plate of 1 mm thickness subjected to a uniform pressure of $q_{z}$ = 1 N/mm${}^{2}$. Each layer is made of the same orthotropic material with the 0${}^{\circ }$/90${}^{\circ }$/0${}^{\circ }$ orientations (for 0${}^{\circ }$, axis 1 corresponds to the global *x* axis). The material properties are presented in Table 4.

In the given example, the FEM23 calculations are performed with the use of a 2D 70 × 70 structural mesh consisting of nine-node quadrilateral elements. The coordinates’ origin is located in the middle of the plate, and the boundary conditions remain the same as presented in Equation (Equation 31). The FEM23 results are compared with the ones yielded by the HR-RZT [31] and 3D FEM [32] methods.

Figure 10 shows the diagrams of two shear stress components $\sigma_{xz}$ and $\sigma_{yz}$ along the plate’s thickness at two selected points of the plate generated with the FEM23, 3D FEM, and HR-RZT, respectively. In this case, the FEM23 results almost coincide with the results of the two other approaches, with only a slight difference observed between them.

## 5. Conclusions

This paper has presented the FEM23 method adapted for a numerical analysis of orthotropic multi-layered plates. The novelty of the paper lies in using the FEM23 approach to derive a numerical model for a single orthotropic layer and extend its application to multi-layered plates. The FEM23 approach utilizes a 2D finite element mesh to perform a full 3D analysis of multi-layered structures. The method, in contrast to the standard 3D FEM, can efficiently deal with structures comprising a combination of thick and thin layers made from different materials. The numerical model generated with the FEM23 is based on the 2D–1D decomposition in the physical model, spatial integration, and approximation. The physical properties of the layers are defined for each one independently. The layers’ orthotropic properties are introduced with the fourth order Hooke’s tensor for orthotropic materials expressed with the local, orthotropic coordinates. Due to the symmetry of Hooke’s tensor, the strain tensor is substituted by the displacement gradient, and the transformation to the global coordinates of the tensor is derived. Then, the numerical models of the individual layers are combined to assemble the global model based on a 2D finite element mesh. After postprocessing, full 3D results are obtained.

The paper is illustrated with examples analyzing multi-layered plates subjected to external loads. In each case, the FEM23 results are compared with the ones taken from several published papers, based on plate theories, or obtained with the standard 3D FEM method. The FEM23 and the 3D FEM results generally coincide. In the case of thin plates, the FEM23 results are comparable with the ones yielded by plate theories. However, when thicker plates are considered, the results begin to diverge, which is common for plate theories. Modelling multi-layered plates is relatively easy in the FEM23 method, where the layers, thick or thin, demonstrating homogenous or orthotropic properties, are combined to assemble the final numerical model.

The full potential of the FEM23 method is yet to be discovered, especially with its Matlab-based software still evolving to be more efficient and able to model a spectrum of mechanical issues. In the near future, the applications of the presented method in the context of multi-layered anisotropic structures may include: buckling and post-buckling analysis, dynamics, thermo-mechanical analysis, or delamination processes using large deformation descriptions.

## Figures and Tables

**Figure 1 materials-14-06959-f001:**
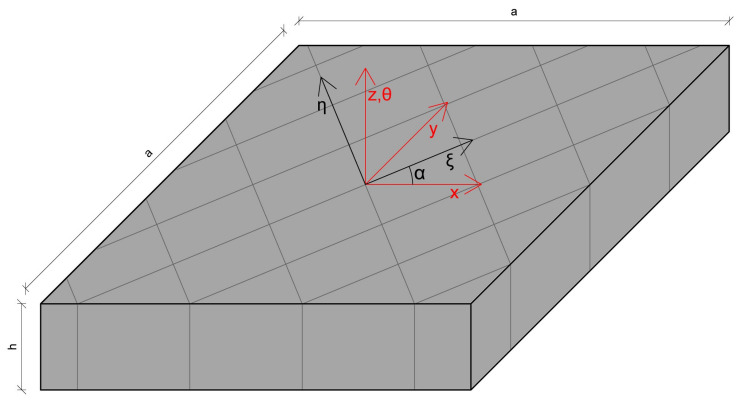
Relation between orthotropic and global coordinates in the plate with orthotropic properties.

**Figure 2 materials-14-06959-f002:**
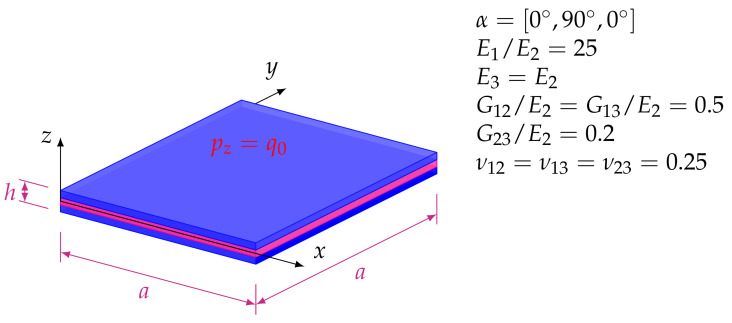
Example 1: Three-layered, simply supported sandwich plate loaded by constant distribution of transverse pressure.

**Figure 3 materials-14-06959-f003:**
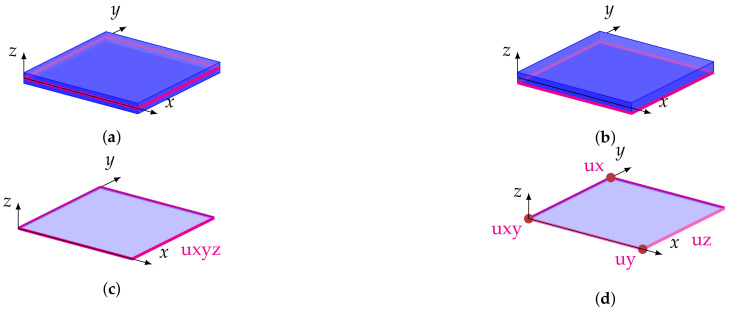
Example 1: Types of boundary conditions for simply supported plate and their symbols. Four versions of boundary conditions: [mh], [ms], [bh], and [bs]. (**a**) boundary condition in the middle of the plate thickness [m{h,s}]; (**b**) boundary condition at the bottom of the plate thickness [b{h,s}]; (**c**) hard simply supported [{m,b}h]; (**d**) soft simply supported [{m,b}s].

**Figure 4 materials-14-06959-f004:**
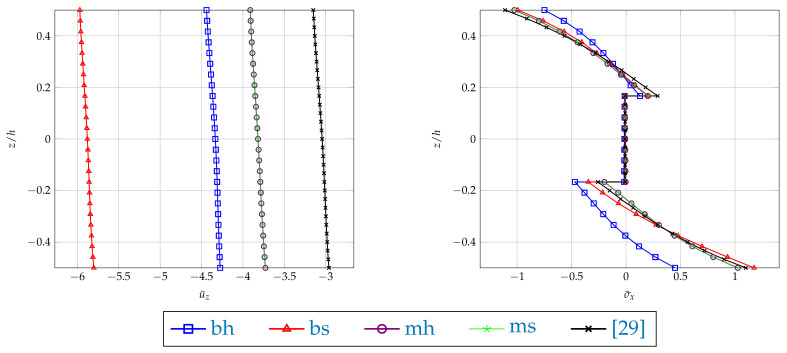
Example 1: Plots of dimensionless deflection ${\bar{u}}_{z}$ and stress component ${\bar{\sigma }}_{x}$ in the middle of the plate with a/h=4, along the plate thickness calculated by FEM23 with different types of boundary conditions.

**Figure 5 materials-14-06959-f005:**
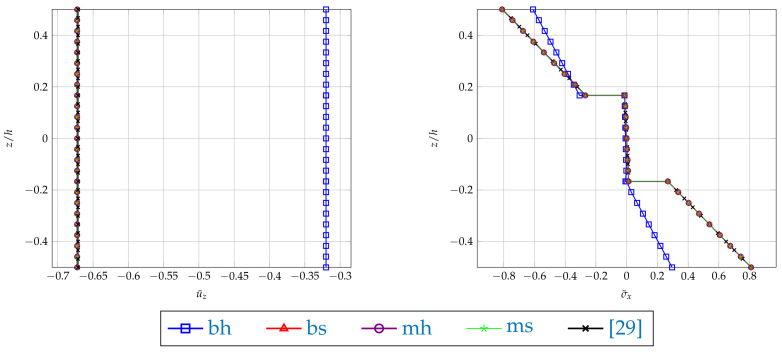
Example 1: Plots of dimensionless deflection ${\bar{u}}_{z}$ and stress component ${\bar{\sigma }}_{x}$ in the middle of the plate with a/h=100, along the plate thickness calculated by FEM23 with different types of boundary conditions.

**Figure 6 materials-14-06959-f006:**
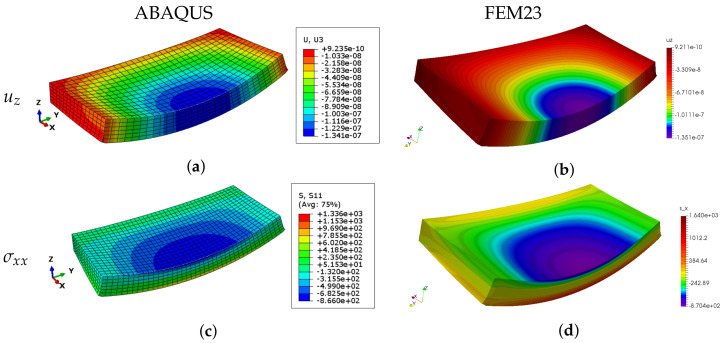
Example 1: The maps of deflection $u_{z}$ (**a**,**b**) and the stress component $\sigma_{xx}$ (**c**,**d**) presented on the half of the deformed plate obtained by 3D FEM and FEM23 for the bs boundary type.

**Figure 7 materials-14-06959-f007:**
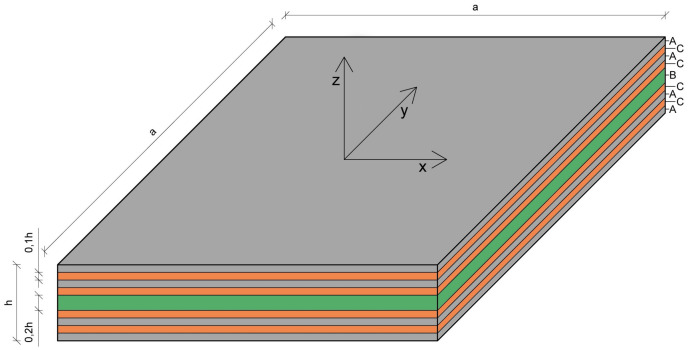
Example 2: Construction of the nine-layered plate.

**Figure 8 materials-14-06959-f008:**
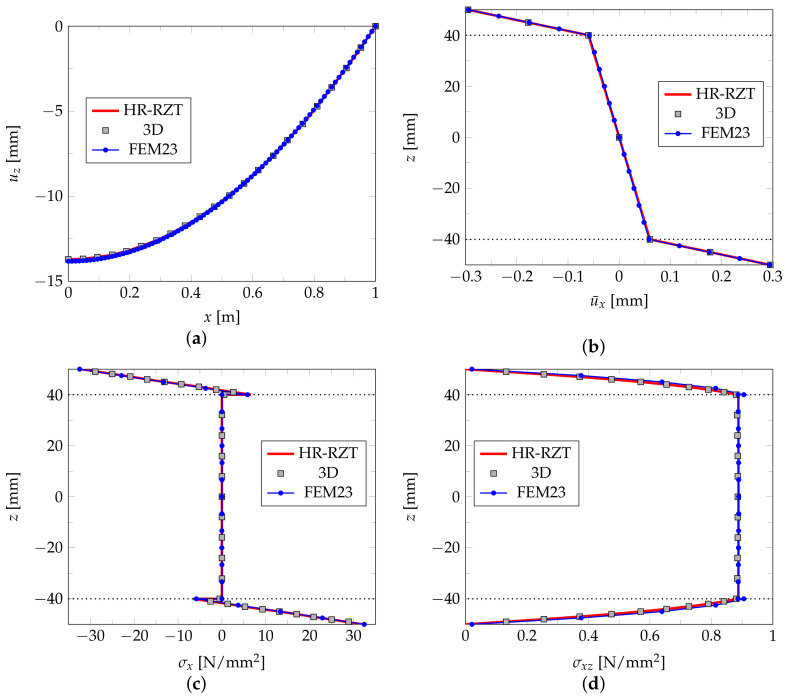
Example 2. Results for plate L1: deflection ${\bar{u}}_{z}\left(x,0,0\right)$ (**a**), displacement ${\bar{u}}_{x}\left(3/7l,0,z\right)$ (**b**), normal stress $\sigma_{x}\left(3/7l,0,z\right)$ (**c**), shear stress $\sigma_{xz}\left(3/7l,0,z\right)$ (**d**).

**Figure 9 materials-14-06959-f009:**
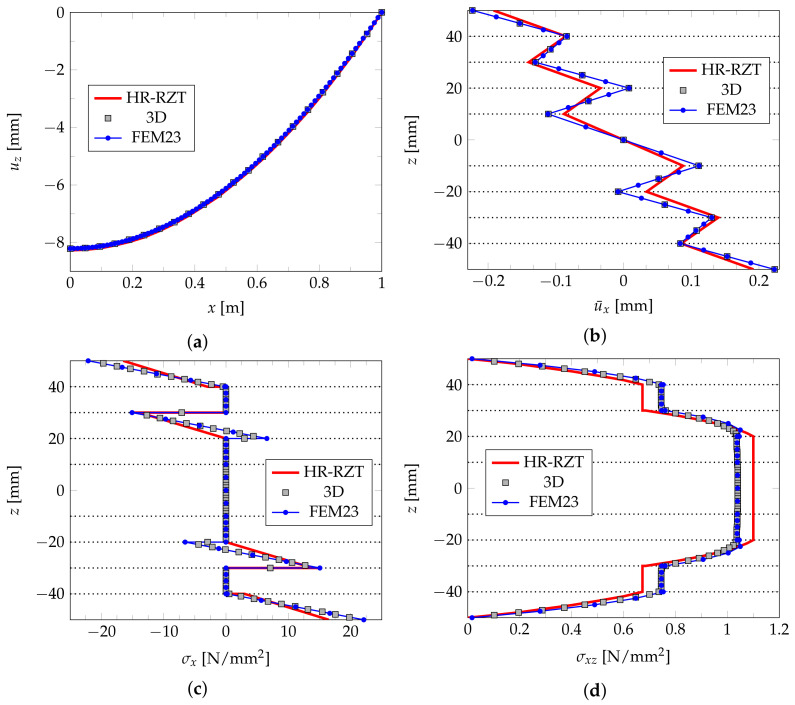
Example 2. Results for plate L2: deflection ${\bar{u}}_{z}\left(x,0,0\right)$ (**a**), displacement ${\bar{u}}_{x}\left(3/7l,0,z\right)$ (**b**), normal stress $\sigma_{x}\left(3/7l,0,z\right)$ (**c**), shear stress $\sigma_{xz}\left(3/7l,0,z\right)$ (**d**).

**Figure 10 materials-14-06959-f010:**
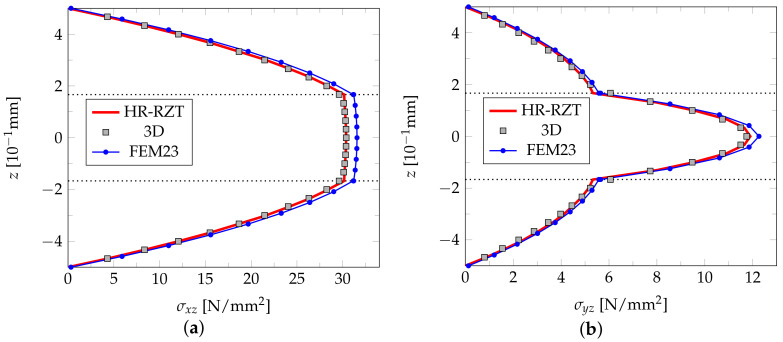
Example 3: shear stress along the plate thickness $\sigma_{xz}\left(3/7l,0,z\right)$ (**a**) and $\sigma_{yz}\left(0,3/7l,z\right)$ (**b**).

**Table 1 materials-14-06959-t001:** Example 1: The results for the simply supported plate.

	Theory		3D Analysis							
a/h	[30]	[29]	FEM23				FEM (Abaqus)			
			mh	ms	bh	bs	mh	ms	bh	bs
	$|{\bar{u}}_{z}|$									
4	3.079	3.044	**3.818**	**3.818**	**4.333**	**5.878**	3.98	3.98	4.04	5.55
10	1.156	1.154	**1.214**	**1.213**	**0.937**	**1.351**	1.22	1.22	0.906	1.34
100	0.671	0.671	**0.678**	**0.672**	**0.316**	**0.672**	0.672	0.672	0.314	0.672
	${\bar{\sigma }}_{x}$									
4	1.116	1.117	**1.026**	**1.028**	**0.450**	**1.175**	1.015	1.016	0.441	1.192
10	0.872	0.871	**0.881**	**0.882**	**0.442**	**0.874**	0.880	0.880	0.417	0.875
100	0.808	0.808	**0.810**	**0.810**	**0.215**	**0.810**	0.810	0.810	0.288	0.810

**Table 2 materials-14-06959-t002:** Example 2: Properties of the materials in the plate.

	A	B	C
$E_{1}^{k}$ [N/mm${}^{2}$]	157,900	191.5	104
$E_{2}^{k}$ [N/mm${}^{2}$]	9584	191.5	
$E_{3}^{k}$ [N/mm${}^{2}$]	9584	1915	
$\mu_{12}^{k}$ [-]	0.32	6.58$\cdot 10^{-3}$	0.3
$\mu_{13}^{k}$ [-]	0.32	6.43$\cdot 10^{-7}$	
$\mu_{23}^{k}$ [-]	0.49	6.43$\cdot 10^{-7}$	
$G_{12}^{k}$ [N/mm${}^{2}$]	5930	4.23$\cdot 10^{-5}$	40
$G_{13}^{k}$ [N/mm${}^{2}$]	5930	365.1	
$G_{23}^{k}$ [N/mm${}^{2}$]	3227	1248	

**Table 3 materials-14-06959-t003:** Example 2: Stacking sequence.

Laminate	Layer Sequence	$h^{k}/h$
L1	A/C/A	0.1/0.8/0.1
L2	A/C/A/C/B/C/A/C/A	0.1/0.1/0.1/0.1/0.2/0.1/0.1/0.1/0.1

**Table 4 materials-14-06959-t004:** Example 3: Material properties.

$E_{1}^{k}$ [N/mm${}^{2}$]	125,000
$E_{2}^{k}$ [N/mm${}^{2}$]	7400
$E_{3}^{k}$ [N/mm${}^{2}$]	7400
$\mu_{12}^{k}$ [-]	0.34
$\mu_{13}^{k}$ [-]	0.34
$\mu_{23}^{k}$ [-]	0.37
$G_{12}^{k}$ [N/mm${}^{2}$]	4800
$G_{13}^{k}$ [N/mm${}^{2}$]	4800
$G_{23}^{k}$ [N/mm${}^{2}$]	2700

## Data Availability

Not applicable.

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
