# Peer review of "Three-Dimensional Bending Analysis of Multi-Layered Orthotropic Plates by Two-Dimensional Numerical Model"

_materials, 2021, doi:10.3390/ma14226959_

Round 1

Reviewer 1 Report

The paper presents FEM23 numerical modelling method to construct the numerical model of multi-layered plates with orthogonal properties. The topic is intersting.   There are only mathematical euquations and simulation results. The main difficulty and main improvement for construct FEM 23 model are missing. It is difficulty to find the innovation points and scientific sounds, therefore, I donot recommend to publish this manuscript.

Author Response

The authors would like to express their  sincere  gratitude to  all the reviewers for evaluation of their paper. The paper has been revised accordingly to the  reviewers’ suggestions. The responses to Reviewers’  comments are provided below.

The paper presents FEM23 numerical modelling method to construct the numerical model of multi-layered plates with orthogonal properties. The topic is intersting.   There are only mathematical euquations and simulation results. The main difficulty and main improvement for construct FEM 23 model are missing. It is difficulty to find the innovation points and scientific sounds, therefore, I donot recommend to publish this manuscript.

The authors would like to thank the reviewer for his  opinion, even though the conclusion is a not  a positiveone . In their paper , the authors proposed  an  alternative approach to the ones known from the field literature, where a  full 3D mechanical model is applied to a  2D numerical one  for a multi-layered plate with orthotropic properties. The numerical methods described in the literature of the field  are usually based on simplified  plate theories. However,  in the presented approach a full 3D analysis is performed  with a relatively simple numerical model on a  2D mesh.  The main contribution of this paper is a demonstration of how easily  any physical relations (in this case,  orthotropic ones ) can be applied in the FEM23 method Due to the anisotropic approximation, the transverse approximation order can be adjusted to the  layer’s thickness, and  consequently, the plates with very thin or thick orthotropic layers can be 3D analyzed.

Reviewer 2 Report

The manuscript entitled "Three-dimensional bending analysis of multi-layered orthotropic plates by two-dimensional numerical model” presented effective numerical modelling of multilayered plates with orthogonal properties. The proposed model was validated by analytical and Finite Element results from the literature.

This reviewer has a big concern about the validation of the developed model using other numerical and analytical results.

Comments:

  • The English writing of the manuscript is not good. Therefore, it could benefit greatly from professional editing to improve technical writing and English.
  • The authors should highlight how their study is providing a different approach or adding significantly to what has been done.
  • Lines 28-31: This sentence should be corrected. What do you mean by "m" in this sentence?
  • Lines 73-74: Is it transverse direction or perpendicular dimension?
  • I think it will be better to provide D as a tensor or matrix. It should be adjusted through the manuscript.
  • The authors should validate the developed model using experimental results. All provided verifications are theoretical or Finite Element results. Also, the authors should highlight what is the difference between the provided three examples. Are there any capabilities that have been highlighted by these examples?
  • Figure 3: I think the question mark in the caption of each figure should be corrected.
  • Table 1: It will be better to provide charts to compare the developed results with those from the literature.
  • Line 136: What do you mean by “good agreement"? How much is this agreement? What are the differences between the results in comparison? This should be addressed throughout the manuscripts.
  • Line 141: It should be Table 2.
  • What is the difference between the first paragraph and the second one of the conclusion section?
  • Line 183: What do you mean by "The model the layers can"?

Author Response

The authors would like to express their  sincere  gratitude to  all the reviewers for evaluation of their paper. The paper has been revised accordingly to the  reviewers’ suggestions. The responses to Reviewers’  comments are provided below.

The manuscript entitled "Three-dimensional bending analysis of multi-layered orthotropic plates by two-dimensional numerical model” presented effective numerical modelling of multilayered plates with orthogonal properties. The proposed model was validated by analytical and Finite Element results from the literature.

This reviewer has a big concern about the validation of the developed model using other numerical and analytical results.

Comments:

The English writing of the manuscript is not good. Therefore, it could benefit greatly from professional editing to improve technical writing and English.

The paper has been thoroughly edited, proof –read and corrected.

The authors should highlight how their study is providing a different approach or adding significantly to what has been done.

The presented approach utilizes  a fully 3D mechanical description in its numerical model.  It can be said that the FEM23 resembles   the approach applied in the solid shells FEM or full 3D FEM. However, in the FEM23 the construction of the final numerical model is  much simpler and the application of some physical relations (in this case, orthotropic ones )  . – much easier . Due to the anisotropic approximation, the transverse approximation order can be adjusted to the layer ‘s thickness, and  consequently,  the plates with very thin or thick layers can be 3D analyzed. The 3D approach for multilayer plates or shell is not very popular in literature of the field  due to numerical problems related to he specific geometry of such structures.  The FEM23 has been proven  to be effective for  the 3D analysis and relatively easy to model multi-layered plates or shells.

The introduction has been modified and our achievements have been highlighted in relation to other solutions in literature.

Lines 28-31: This sentence should be corrected. What do you mean by "m" in this sentence?

Authors agree with the reviewer, that the sentence is unclear. This sentence has been corrected as follows:

In previous papers [9,14] , the authors have already utilized the FEM23 to model the behavior of multi-layered plates and shells with isotropic layers subjected to mechanical or thermo-mechanical loads, devoting special attention to laminated glass structures-   the topic of many numerical or experimental studies.[13,18,23,24].

The “m” was just an editing misprint.

Lines 73-74: Is it transverse direction or perpendicular dimension?

The paragraph   has been modified, to clarify the message

Two Cartesian coordinates : the orthotropic 100 (ξ, η, θ) and global (x, y, z) ones are distinguished in the orthotropic plate. The θ and z directions are the same and oriented transversely  , while both the (ξ, η) pair , associated with the in-plane orthotropy in a layer,  and the (x, y) pair are correlated with the in-plane geometry of the plate. Although the (x, y) pair is common for the 104 whole plate, the (ξ, η) directions are oriented differently in each layer.

I think it will be better to provide D as a tensor or matrix. It should be adjusted through the manuscript.

Authors have chosen  to use the full tensor notation, in which the Hooke’s constitutive tensor D is defined as the 4th order one, and strains and stresses are expressed  with  the symmetric 2nd order tensors. Such a notation allows for a clear, elegant and compact formulation of a problem.  Expressing such a relation as the one in the equation (4)  would be problematic with the Voigt notation, and completely impossible in the case of (11)-(15) equations, where the displacement gradient,  instead of the strain tensor,  is used.. The computer application of the FEM23 is  prepared in Matlab with  the full tensor algebra, exactly  in the manner presented   in this paper.

The authors should validate the developed model using experimental results. All provided verifications are theoretical or Finite Element results. Also, the authors should highlight what is the difference between the provided three examples. Are there any capabilities that have been highlighted by these examples?

The authors fully agree with the reviewer that each new method or approach should finally be verified using experimental data. The authors are going to to provide  such a validation by  reference to the already  published   results , , for instance

  1. Battaglia, A. Di Matteo, G. Micale, A. Pirrotta, Vibration-based identification of mechanical properties of orthotropic arbitrarily shaped plates: Numerical and experimental assessment, Composites Part B: Engineering, Volume 150, 2018, Pages 212-225.

However, experimental results might be  ‘contaminated ’ by some unpredicted effects, such as: fthe ones associated with fixing or supporting the given specimen, material properties uncertainty, measurement errors, etc.

It is a common situation in  literature that  any new methods are  compared with other known ones  for validation purposes, verifying at the same time the accuracy  of the mathematical formulae derived by the new method, see the references presented in this study The same approach has been adopted  in this paper, where the FEM23 has been  compared with  other 2D or 3D methods, where various theories have been applied. The examples presented in this paper show good correlations with other methods, which confirms the accuracy  of the mathematical formulae of the FEM23.

The experiments concerning  the orthotropic plates focus  usually on the structural dynamical response  expressed in the form of  free-vibrations. Using the FEM23 method for the dynamic problem would require a reformulation of the approach in order to conduct  comparison with the experimental results. That  will  constitute  the subject  of our further research.

The preamble to Section 4 (Examples) has been modified in order  to highlight the  capabilities of the FEM23  and differences observed in the examples.

Figure 3: I think the question mark in the caption of each figure should be corrected.

The captions have been corrected.

Table 1: It will be better to provide charts to compare the developed results with those from the literature.

The Figures 4 and 5 have been modified in such a way that the FEM23 results are compared with the ones based on the plate theory presented in [4]. The results coincide  when a thin plate is considered  , and demonstrate slight  differences in the case of  a thick plate. Such an effect is typical,  for plate theory approaches which generally, do not yield good results when the plate is thick.

Line 136: What do you mean by “good agreement"? How much is this agreement? What are the differences between the results in comparison? This should be addressed throughout the manuscripts.

The comment has been modified.

Line 141: It should be Table 2.

Corrected

What is the difference between the first paragraph and the second one of the conclusion section?

Line 183: What do you mean by "The model the layers can"?

The whole ‘Conclusions’ section has been rewritten.

Reviewer 3 Report

In the current research study, the FEM23 method previously presented by the authors is extended to analyze a multi-layered plate with orthotropic properties. The topic under consideration is very interesting and worthy of investigation. The manuscript reads well and the write-up is clear making it easy to read. However, from the reviewer’s point of view, the scientific quality of the manuscript appears to be more of a case study and advanced engineering rather than cutting-edge research. The focus and discussions are limited and there is no attempt to look into the underlying physics of the problem. It is expected that the authors could provide more insight into the behavior of the plated structures, based on more advanced experimental methods (e.g. optical methods like 3D-DIC), theoretical and computational models available within the current state of the art. The complexity in measuring the full-field deformations and modelling the loading in these kinds of environments is widely acknowledged, but this kind of insight would significantly strengthen the manuscript and provide more knowledge on how multi-layered behave in these loading environments. Thus, this work fails to provide fundamental insights into the problem under consideration and should seek to improve the current version of the manuscript. In addition to the already mentioned comments, the authors are recommended to consider the following to improve the current version of the manuscript:

1) The range of this study is very narrow. There exists considerable work on the topic of multi-layered plates. The authors should therefore consider including a broader scope of the work on blast-loaded plates in their introduction, to better communicate the limitations of this study. The following papers can be helpful:

(The introduction needs to be improved by mentioning some other papers related to the topic.)

[1] Ziya-Shamami, Mojtaba, et al. "Structural response of monolithic and multi-layered circular metallic plates under repeated uniformly distributed impulsive loading: An experimental study." Thin-Walled Structures 157 (2020): 107024.

2) The authors should also provide more insight into the underlying physics of the problem. Is the method can be extended for plated structures under repeated loading or extreme dynamic loading? If yes, please compare your results with the above-mentioned papers.

3) How the interaction between the layers was considered in the model?

4) The concluding remarks appear as a rather trivial. Please provide your future outlook and applications.

Author Response

The authors would like to express their  sincere  gratitude to  all the reviewers for evaluation of their paper. The paper has been revised accordingly to the  reviewers’ suggestions. The responses to Reviewers’  comments are provided below.

In the current research study, the FEM23 method previously presented by the authors is extended to analyze a multi-layered plate with orthotropic properties. The topic under consideration is very interesting and worthy of investigation. The manuscript reads well and the write-up is clear making it easy to read. However, from the reviewer’s point of view, the scientific quality of the manuscript appears to be more of a case study and advanced engineering rather than cutting-edge research. The focus and discussions are limited and there is no attempt to look into the underlying physics of the problem. It is expected that the authors could provide more insight into the behavior of the plated structures, based on more advanced experimental methods (e.g. optical methods like 3D-DIC), theoretical and computational models available within the current state of the art. The complexity in measuring the full-field deformations and modelling the loading in these kinds of environments is widely acknowledged, but this kind of insight would significantly strengthen the manuscript and provide more knowledge on how multi-layered behave in these loading environments. Thus, this work fails to provide fundamental insights into the problem under consideration and should seek to improve the current version of the manuscript. In addition to the already mentioned comments, the authors are recommended to consider the following to improve the current version of the manuscript:

1) The range of this study is very narrow. There exists considerable work on the topic of multi-layered plates. The authors should therefore consider including a broader scope of the work on blast-loaded plates in their introduction, to better communicate the limitations of this study. The following papers can be helpful:

(The introduction needs to be improved by mentioning some other papers related to the topic.)

[1] Ziya-Shamami, Mojtaba, et al. "Structural response of monolithic and multi-layered circular metallic plates under repeated uniformly distributed impulsive loading: An experimental study." Thin-Walled Structures 157 (2020): 107024.

The authors fully agree with the reviewer.Experimental studies  on multi-layered and orthotropic plates have been included in the introduction.

2) The authors should also provide more insight into the underlying physics of the problem. Is the method can be extended for plated structures under repeated loading or extreme dynamic loading? If yes, please compare your results with the above-mentioned papers.

The FEM23 provides an  alternative way, beside   the standard 3D FEM, to construct a  numerical model of a considered problem in the multi-layered domain. This method can be used for various types of problems.  As mentioned in the revised conclusions the dynamic analysis (including the dynamic loading) is probably going to be applied in the FEM23 in the  nearest  future.

3) How the interaction between the layers was considered in the model?

In this particular paper it is assumed that the displacements are continuous at the layers’ boundaries. The jumps of tangent stresses and continuity of the normal stress to the inter-layer surface come naturally in this method. Other types  of interactions apply when delaminations are modelled. Then,  displacement discontinuities are allowed in the form of normal openings or inter-layers sliding. However,  this will be the subject     of future  articles. In the FEM23,  the bonding layers, which are usually very thin, are also modelled as  3D layers.

4) The concluding remarks appear as a rather trivial. Please provide your future outlook and applications.

The concluding remarks have been fully rewritten and include  future applications.

Reviewer 4 Report

A comprehensive study has been performed to analyze the response of orthogonal multi-layered plates using a novel method. The paper was well organized and developed. There is no further comment.

The obtained results have been interpreted properly and all conclusions are supported by them. Besides, the quality of presentation and analysis as well as the scientific soundness are well. Moreover, the methods, tools, software, and reagents have been described with sufficient details to allow another researcher to reproduce the results.

Author Response

The authors would like to express their  sincere  gratitude to  all the reviewers for evaluation of their paper. The paper has been revised accordingly to the  reviewers’ suggestions. The responses to Reviewers’  comments are provided below.

A comprehensive study has been performed to analyze the response of orthogonal multi-layered plates using a novel method. The paper was well organized and developed. There is no further comment.

The obtained results have been interpreted properly and all conclusions are supported by them. Besides, the quality of presentation and analysis as well as the scientific soundness are well. Moreover, the methods, tools, software, and reagents have been described with sufficient details to allow another researcher to reproduce the results.

The authors would like to thank the reviewer for  his  opinion.

Round 2

Reviewer 1 Report

The paper can be accepted in present form.

Reviewer 2 Report

The authors have addressed most of the reviewer's comments and the manuscript can be accepted for publication.

Reviewer 3 Report

My concerned questions have been addressed properly in the response letter to the Editor and in the revised manuscript. Therefore, I recommend the manuscript be published.